# Molecular Landscape of Pediatric Thyroid Cancer: A Review

**DOI:** 10.3390/diagnostics12123136

**Published:** 2022-12-12

**Authors:** Prerna Guleria, Radhika Srinivasan, Chanchal Rana, Shipra Agarwal

**Affiliations:** 1Department of Pathology, Command Hospital (Southern Command), Pune 411040, India; 2Department of Cytology & Gynecological Pathology, Post Graduate Institute of Medical Education and Research, Chandigarh 160012, India; 3Department of Pathology, King George’s Medical University, Lucknow 226003, India; 4Department of Pathology, All India Institute of Medical Sciences, New Delhi 110029, India

**Keywords:** pediatric thyroid cancer, molecular, somatic mutations, fusions, familial, miRNA, thyroid differentiation genes

## Abstract

Thyroid carcinomas (TC) are rare in the pediatric population; however, they constitute the most common endocrine malignancy. Despite some similarities with adult carcinomas, they have distinct clinical behavior and responses to therapy due to their unique pathology and molecular characteristics. The age cut-off used for defining the pediatric age group has been variable across different studies, and the universally accepted recommendations influence accurate interpretation of the available data. Moreover, factors such as radiation exposure and germline mutations have greater impact in children than in adults. Papillary TC is the most common and the most evaluated pediatric TC. Others, including follicular, poorly differentiated and medullary carcinomas, are rarer and have limited available literature. Most studies are from the West. Asian studies are primarily from Japan, with few from China, India, Saudi Arabia and Republic of Korea. This review provides a comprehensive account of the well-established and novel biomarkers in the field, including point mutations, fusions, miRNA, and thyroid differentiation genes. Familial and syndromic associations are also discussed. Current management guidelines for pediatric patients are largely derived from those for adults. An awareness of the molecular landscape is essential to acknowledge the uniqueness of these tumors and establish specific diagnostic and therapeutic guidelines.

## 1. Introduction

Thyroid malignancies commonly arise from follicular cells and encompass differentiated thyroid carcinoma (DTC), poorly differentiated thyroid carcinoma (PDTC), and anaplastic thyroid carcinoma (ATC). DTCs maintain the normal physiologic characteristics of thyroid follicular cells [1], and papillary thyroid carcinoma (PTC) forms the bulk (>90%), followed by follicular thyroid carcinoma (FTC) (<10%) [2]. Medullary thyroid carcinoma (MTC) is another subset of thyroid tumors which arises from the parafollicular cells [2].

Though rare, thyroid malignancy is the most common endocrine malignancy in the pediatric age group [3,4]. The World Health Organization considers 19 years the age cut-off for segregating the pediatric population from adults. Subjects younger than 9 years old are considered children, and subjects of 10–19 years are considered adolescents [5]. The American Thyroid Association (ATA) takes 18 years as the cut-off [6,7], whereas the American Academy of Pediatrics identified the upper age limit as 21 years [8]. There is, to date, no consensus on the age cut-off to be used to define the pediatric age group for thyroid malignancies; the upper limit varies from 18 years [9,10] to 22 years [11,12] in different studies. A recent Japanese study recommended 14 years as the cut-off. The authors found better disease-free survival and distant metastasis-free survival in DTC patients aged <15 years [13]. 

In contrast to adults, thyroid nodules are rarer (1–3%) in the pediatric age group, but when present, are more likely to be malignant (19–26%) [14,15], with the peak incidence of malignancy being among 15–19 years old. Fortunately, despite an advanced stage of disease at presentation and higher recurrence rates, the mortality rate remains low (<2%) [4,6,15,16]. Pediatric patients are also more likely to respond to radioactive iodine therapy (RAI) than adults [17]. This has been partly ascribed to the distinct molecular landscape of pediatric tumors.

Owing to the apparent differences in the clinical behavior and pathophysiology of thyroid cancer involving the pediatric age group, the ATA has laid down separate recommendations for the management of these patients [6]. The molecular makeup of these tumors has been variably evaluated. This review provides a comprehensive synopsis of the molecular profile of thyroid cancers occurring in pediatric patients, along with a comparison with adults, while focusing on the Asian literature. 

## 2. Epidemiology

Thyroid malignancy is rare in the pediatric age group. The recent Surveillance, Epidemiology, and End Results (SEER) database revealed an incidence of only 1.9% of all cancers in patients less than 20 years of age [3]. As is the case in adults, PTC is the most common histological type (80–90%), followed by FTC (5–10%) [18,19]. Other primary thyroid carcinomas, such as MTC (3–5%), PDTC and ATC are even rarer. Most patients present in the second decade, with FTC showing a predisposition for slightly older patients than PTC [17]. There is a female preponderance. The incidence rates in males and females are 0.2 and 0.6 per 1,000,000 in children aged 0–14 years, and 1.2 and 6 per 1,000,000 in the age group 15–19 years [3]. Hence, the differences in the incidence rates in males and females are more pronounced in the post-pubertal age group.

As per the GLOBOCAN (Global cancer observatory) data 2020, Asia contributes more than half (56%) of total new thyroid carcinoma cases in the pediatric age group (0–19 years). Considering individual countries, China (23.2%) supersedes all, followed by the United States of America (9.2%) and India (8.6%) (Figure 1) [20]. The data are influenced by the size of the population, environmental factors and the disease surveillance system specific to the country. The presence of mixed ethnicity in countries such as the United States of America also plays an essential part in determining disease epidemiology.

## 3. Risk Factors

Most pediatric DTCs are sporadic [21]. Exposure to radiation/radiotherapy, autoimmune thyroid disorders, iodine deficiency and familial/genetic syndromes are the common predisposing factors.

### 3.1. Exposure to Radiation/Radiotherapy

Radiation-induced thyroid malignancies occur at an early age and are more often multifocal [1]. Radiation exposure can be external (head and neck radiotherapy) or internal (radioiodine intake post-nuclear plant accidents, such as Chernobyl). The carcinogenic effect of radiation has been seen with exposures of >50 cGy, with up to 17% incidence of DTC following exposure. The incidence of thyroid malignancies increased dramatically in Belarus, Ukraine and Russia within four years of the Chernobyl accident when compared with a decade before that [22,23,24,25]. Radiation exposure is associated with genetic alterations that are distinct from children with sporadic PTC. Studies conducted on post-Chernobyl PTC patients have reported a higher prevalence of *RET*/*PTC3* (55–60%) translocations than *RET*/*PTC1* (15–25%). A reverse pattern is seen in sporadic cases (15–20%, and 45–55%, respectively) [26,27,28]. New fusion partners of *RET*, namely *PTC5*, *PTC6*, *PTC7*, *RFGX*, *RFGX*, *AFAP1L2* and *PPFIBP2* not otherwise reported in sporadic cases, have also been found in post-radiation PTCs [28,29,30]. The difference in the molecular profile is also reflected in the histopathological subtype of PTC seen in these patients. Nikiforov et al. reported a higher prevalence of the solid subtype (37%) in radiation-induced PTC compared with sporadic PTCs in children (4%) [26].

Additional molecular alterations have been reported in post-radiation thyroid carcinoma. In a study by Nikiforov, three of the 17 cases (18%) of pediatric post-Chernobyl PTCs revealed the presence of minisatellite instability, which was absent in the 20 sporadic cases tested. While minisatellites are segments of repetitive DNA 10–100 bp long, microsatellites are shorter, at 1–9 bp long. The authors found microsatellite instability in only one (6%) tumor. Hence, they suggested the involvement of somatic minisatellite mutations in the pathogenesis of radiation-induced thyroid tumorigenesis [22].

### 3.2. Autoimmune Thyroiditis

The association of thyroid malignancy with autoimmune thyroiditis has been controversial, with some even hypothesizing a favorable clinical outcome due to the protective environment provided by infiltrating lymphocytes [31,32,33,34]. Corrias et al. examined this association in a large multicentric cohort from Italy, and reported a prevalence of thyroid nodules and thyroid cancer in 31.5% and 3%, respectively; PTC was the only histotype detected [35]. Sur et al., in their recent review, observed that 3.07% of the patients with Hashimoto thyroiditis developed PTC over 2–10 years. [36]. There are studies with contrasting results, too. A study from China evaluated clinicopathological features of pediatric patients of DTC < 18 years of age; while 44.2% of these had coexistent HT, a similar percentage (41.3%) had nodular goiter [37]. Overall, PTCs with background HT are more likely to be multifocal, but show favorable histological features. Increased production of thyroid-stimulating hormone and the chronic inflammatory infiltrate with the ensuing cellular proliferation, increased angiogenesis and reduced apoptosis are contributing factors [38]. Subhi et al. analyzed the microarray expression profiles of cases of HT and those of PTC with HT. They found upregulation in the number of immunoglobulin kappa variable genes, as well as other immune-related genes, including those associated with oxidative stress, reactive oxygen species, DNA damage, DNA repair, cell cycle and apoptosis [39]. 

### 3.3. Iodine Content

The association of thyroid cancers with iodine content in the body has been controversial. Earlier reports indicated an increased incidence with iodine-deficient status, whereas later research has shown a reverse association. Studies on rat models reported both iodine deficiency and iodine excess acted as tumor promoters. However, there is no conclusive proof of the same in humans. Moreover, the distribution of iodine varies in different geographic regions, precluding actual measurement of the iodine intake and its content in food items. Interestingly, research has demonstrated an association of different thyroid cancer subtypes with the iodine status of the individual, with FTC and ATC being more common in deficiency, and PTC in excess [40,41,42]. 

### 3.4. Familial/Genetic Syndromes

Familial thyroid cancers can arise from C-cells or follicular cells. While the former is more common and leads to the development of MTC, the latter are called ‘familial non-medullary thyroid carcinomas’ (FNMTC) [43]. FNMTCs form 5–15% of all thyroid malignancies and can be syndromic or non-syndromic [44]. The syndromic ones include familial adenomatous polyposis (FAP), Cowden syndrome, Werner syndrome, Carney complex, DICER1-pleuropulmonary blastoma familial tumor predisposition syndrome and Pendred syndrome. Patients with FAP have a predisposition to developing the cribriform morular thyroid carcinoma with a prevalence of 2–12%. Besides the germline *adenomatous polyposis coli* (*APC*) mutation, the tumors show additional somatic single or multiple molecular alterations in *APC*, *CTNNB1*, *RET*/*PTC*, or *RAS* [1,45]. Patients with Cowden syndrome have follicular neoplasms in up to 10% of patients with only co-incidental detection of PTC. In the Carney complex, caused by mutations in the *protein kinase A regulatory subunit type Ia* gene (*PRKAR1A*), both PTCs and follicular neoplasms may be found. Pendred syndrome results from mutations in the *SLC26A4* (*PDS*) gene, which encodes for the protein pendrin. It is characterized by the triad of bilateral sensorineural deafness, mutism, and goiter. The development of thyroid carcinoma is rare and usually related to chronic stimulation by thyroid-stimulating hormone. Those with Werner syndrome harbor the *WRN* gene mutation and have an increased risk of developing PTC, FTC, and ATC [46]. *DICER1* mutations have now been identified as important drivers in pediatric thyroid nodules, with an overall reported rate of 30%, in contrast to about 1% in adults [47]. These may be germline or somatic, and phenotypic presentation may be in the form of benign or malignant thyroid disease. The disease is usually multifocal and nodular, ranging from adenomatous goiter to true neoplasms. The latter include FA, PTC, FTC, PDTC, and the very rare carcinosarcoma, and malignant teratoma [48,49,50,51,52,53]. While macrofollicular architecture is associated with somatic alterations [50], co-occurrence of DTC and Sertoli–Leydig cell tumor has been suggested to be highly indicative of DICER1 syndrome [54]. DTCs have also been reported in other syndromes including Beckwith–Wiedemann syndrome, Li–Fraumeni syndrome, familial paraganglioma syndromes, McCune–Albright syndrome, and Peutz–Jeghers syndrome [6]. Non-syndromic diseases with a preponderance of NMTC include PTC associated with papillary renal cell neoplasia (PTC-PRN), familial multinodular goiter with PTC (MNG-PTC), familial PTC (fPTC) and familial thyroid carcinoma with and without oxyphilia (TCO). Genetic susceptibility to the development of the above familial tumors has been attributed to six potential regions harboring the following genes: *MNG1* (14q32), *TCO* (19p13.2), *fPTC/PTC-PRN* (1q21), *NMTC1* (2q21), *FTEN* (8p23.1–p22), and the telomere–telomerase complex [44]. The *BRAF* V600E mutation, commonly found in sporadic PTCs, is not seen in familial cases [44]. Furthermore, FNMTC can also be found in isolation, without any identifiable susceptibility loci and in the absence of the commonly found mutations in sporadic DTCs. Although ATA has not recommended the screening of family members of these patients, some authors suggest screening with ultrasonography, if three or more family members are affected, starting from the age of 20, or 10 years before the earliest age of diagnosis in the family [55]. Approximately 10–20% of MTCs are familial [56], developing in multiple endocrine neoplasia (MEN) 2A syndrome (Sipple’s syndrome), MEN 2B syndrome, and familial MTC, all having *RET* mutations as driver alteration [57]. 

## 4. Molecular Profile of Differentiated Thyroid Carcinomas

The mutational landscape of DTCs involves somatic point mutations of *BRAF* and *RAS* genes, and fusions involving the *RET* and *NTRK1* tyrosine kinases. There is consequent activation of the mitogen-activated protein kinase (MAPK) and phosphoinositide 3-kinase (PI3K) signaling pathways. Point mutations are commonly seen in adults (~70%), but are less frequent in children (~30%); instead, gene fusions, which occur at a lower rate in adults (~15%), predominate (~50%) [1]. 

### 4.1. Papillary Thyroid Carcinoma

PTCs account for about 90% of all childhood thyroid cancers. The classic and the follicular subtypes of PTC, considered ‘low-risk’ in adults, are also the most common histological types encountered in this age group. As most of the studies on pediatric thyroid neoplasms are from the pre-noninvasive follicular thyroid neoplasm with papillary-like nuclear features (NIFTP) period, its exact proportion is unknown. Limited evidence suggests that NIFTP constitutes about 4.5% of all PTC cases [58]. About 15–40% of pediatric PTCs are subtypes categorized as ‘high-risk’ in adults, namely tall cell, diffuse sclerosing, and solid/trabecular subtypes [14,59]. Although there are limited follow-up data, there is a suggestion that some of these ‘high-risk’ subtypes tend not to have a worse rate of event-free survival in the pediatric population [60,61].

In contrast to those in adults, PTCs in children are more commonly multifocal and have a more aggressive presentation, with higher rates of distant and nodal metastases. Despite a higher recurrence rate and more aggressive clinical presentation, their prognosis is excellent, with a very low mortality rate [6]. These differences are probably related to significant differences in the underlying molecular genetics of pediatric compared with adult DTC [62]. In addition, a decrease in the male-to-female incidence ratio in post-puberty indicates that other factors, such as different endocrine, metabolic and immune characteristics of the pediatric age, may also be involved. Being the predominant thyroid cancer type in the pediatric population, PTC has been studied relatively more than other thyroid cancers for molecular drivers.

In adult-onset PTCs, the most frequent genetic alteration observed is the mutational activation of the *BRAF* oncogene. A transversion of thymidine to adenine (T1799A) results in the substitution of valine to glutamate at residue 600 (V600E). Of the 402 cases of PTC evaluated in the Cancer Genome Atlas (TCGA) study, 58.5% harbored *BRAF* V600E mutation. Most patients were adults, except for nine patients aged < 20 years. Only 22% of the tumors developing in the latter group showed this mutation [63]. Although radiation exposure has a bearing on oncogenic molecular events, overall, *BRAF* V600E point mutation is less common in pediatric cases. Most of the available literature is from the West, with limited data from Asian countries, to which the largest contribution is from Japan [64]. In their study from Japan, Oishi et al. showed a higher prevalence of *BRAF* V600E in adult PTCs (85%) than in pediatric patients (54%). They also documented a greater frequency in patients aged 16–20 years (62%) compared with those < 15 years (28%) [65]. Other studies have also documented an increased prevalence of *BRAF* V600E mutation in patients of an age > 15 years. Sporadic cases tend to show a higher frequency (0–63%) of *BRAF* V600E mutation than those developing post radiation exposure (0–8%) [62,64]. There is high variability in the reported frequency of the mutation, with one series reporting it to be as high as 63% [66]. The reason for this may be the higher cut-off age of 22 years used [67]. Recently, Mitsutake et al. evaluated the genetic profile of PTCs detected during a survey following the Fukushima Daiichi nuclear reactor accident, and, unlike post-Chernobyl PTCs, found a higher prevalence of *BRAF* V600E [68]. 

In adults, *BRAF* mutation has been suggested to be a poor prognostic factor contributing to progressive disease and poor response to therapy [69]. This correlation remains unconfirmed in pediatric patients, but most of the available data suggest a lack of any such association [65,70]. Recently, Chakraborty from India evaluated 98 pediatric PTC patients for *BRAF* V600E mutation using Sanger sequencing, and found a prevalence of 14.3%. Their study cohort included 68 patients aged ≤ 18 years and 30 patients aged 19–20 years of age. This multivariate analysis revealed RAI refractoriness to be significantly associated with *BRAF* V600E mutation. However, none of the 17 patients with distant metastases had *BRAF* V600E mutation, and there was a lack of any significant association of *BRAF* V600E mutation with the status of disease recurrence or progression [71]. Contrasting results were reported by Alzahrani, who found persistent/recurrent disease to be significantly more common in patients with *BRAF* V600E mutation than in those without [7]. However, a subsequent study by the same group on a larger cohort did not find any association of *BRAF* V600E with any of the aggressive clinicopathological features, including persistent/recurrent disease [72].

Besides point mutations, the TCGA study reported *BRAF* fusions in 2.7% of their PTC cases. There is enough evidence documenting a higher prevalence of *BRAF* fusions in the pediatric age. The most common ones include *Acylglycerol kinase* (*AGK*)/*BRAF* and *A-kinase anchoring protein 9* (*AKAP9*)/*BRAF*. Both fusions result from paracentric inversions involving chromosome 7; inv (7)(q34), and inv(7)(q21q34), respectively. Initially identified in post-radiation-exposed individuals, the fusions have been found even in sporadic pediatric cases [73]. Their prevalence is especially high in younger patients < 10 years of age [73,74]. While the reported frequency of *AKAP9*/*BRAF* has ranged from 0–1%, and 0–11% in sporadic and post-Chernobyl tumors, *AGK*/*BRAF* occurs at a rate of 0–19%, and 0–4%, respectively [73]. Cordioli et al., in two separate studies from their institute, documented for the first time the presence of *AGK*/*BRAF* in 10% and 19% of their sporadic pediatric PTC cases, respectively [74,75]. There is limited evidence suggesting an association of *AGK*/*BRAF* fusion with younger age and distant metastasis [75]. Interestingly, *AGK*/*BRAF* shows geographic variation in distribution [76], being more common in Brazil than in the United States or the Czech Republic [73]. Novel *BRAF* fusion partners identified in some geographical regions of the world include *OPTN*, *CUL1* (Czech Republic) and *SND1*, *MACF*, *MBP*, *POR*, and *ZBTB8A* (post-Chernobyl Ukrainian-American patients) [73]. Pekova et al. studied novel fusion genes in 93 pediatric PTC patients up to 20 years of age, of which 30 had a family history of thyroid disorder. They found 20 different types of fusion genes in 56% of patients, and 5 were novel. Fusion gene-positive cases were associated with aggressive disease, more frequent extrathyroidal extension, and lymph node and distant metastases, and also required higher doses of RAI treatment [77]. The Ukrainian-American population studied by Efanov et al. included 65 PTCs developing in patients < 18 years of age post exposure to radiation during the Chernobyl accident. Gene fusions were observed in 46 patients, including novel fusions, as described above [78].

*RET* (rearranged during transfection) is absent in the normal thyroid follicular cells. It has approximately 20 fusion partners, of which *RET*/*PTC* is the most commonly associated with both sporadic and radiation-induced PTC [1,79]. *RET*/*PTC* rearrangements were found in 6.3% of the PTC tumors included in the TCGA cohort, but were much more frequent (22%) in their pediatric cohort [63]. Interestingly, these represent the most common molecular alterations encountered in children and adolescents [76], both in sporadic cases (22–65%) and tumors developing after radiation exposure (33–77%) [62,73]. *RET*/*PTC1* and *RET*/*PTC3* are the most common rearrangements. Among these, *RET*/*PTC3* is associated with more aggressive disease [1,73]. Classic, solid, and diffuse sclerosing PTC histotypes, and the aggressive clinicopathological parameters such as extrathyroidal extension, lymph node and distant metastases are more commonly associated with *RET* fusions [1]. The prevalence of *RET*/*PTC* and *BRAF* V600E mutations varies with age and ethnicity. While *RET*/*PTC* fusions are more common in Caucasian children < 15 years of age, *BRAF* V600E is more common in the older Hispanic population [67].

Point mutations in the *RAS* genes (*HRAS*, *NRAS* and *KRAS*) are found in up to 25% of PTC cases, particularly in the follicular subtype [80]. 12% of the PTC cases included in the TCGA cohort harbored *RAS* mutations [63]. The incidence is lesser in the pediatric age group (<10%). Codon 61 of the *NRAS* gene is the most commonly involved, and as in adults, there is an association with the follicular subtype [1,72]. Kumagai from Japan found *RAS* mutations in two of their 77 cases (2.6%) of PTC involving children, adolescents and young adults. None of the patients aged < 15 years harbored this mutation [81]. Alzahrani also reported a low frequency of 2.5% in 79 PTC patients < 18 years of age [72]. Mitsutake did not find these in any of their 67 PTC cases [68]. 

*TERT* C288T and C250T promoter mutations occur in 10–20% of adult DTCs [43]. These were present in 9.4% of the cases of the TCGA cohort. None of their pediatric patients had this alteration [63]. In a recent study from India, none of the 98 patients harbored *TERT* promoter mutations [71]. In another study based on 81 sporadic pediatric PTC patients from Japan, *TERT* promoter mutations were absent in all [65]. Using NGS, Franco found *TERT* C288T mutation in a single case of PTC (follicular subtype) out of 29 PTC cases [82]. Most other authors have also observed low frequencies [73,83]. A single study from China has documented a higher prevalence of *TERT* promoter mutations. Geng observed *TERT* C228T mutation in 27% of their 48 PTC patients. The molecular alterations significantly correlated with aggressive clinicopathological features. None of their cases had the C250T mutation [84].

Additional oncogenic mutations associated with pediatric PTCs are *PAX8*/*PPARG* and the *NTRK1* and *NTRK3* fusions; however, data are limited. Using ThyGeNEXT, an NGS panel for detecting mutations in *ALK*, *BRAF*, *GNAS*, *HRAS*, *KRAS*, *NRAS*, *PIK3CA*, *PTEN*, *RET*, or *TERT*, and 38 fusion transcripts involving oncogenes *ALK*, *BRAF*, *NTRK*, or *RET*, Franco documented *STRN*/*ALK*, *ETV6*/*NTRK3*, and *PAX8*/*PPARG*, each in 6.9% (2/29) of their PTC cases [82]. Of the nine pediatric tumors included in the TCGA cohort, *ETV6*/*NTRK3* was found in a single patient and *PAX8*/*PPARG* in none [63]. The *STRN*/*ALK* rearrangement, though rare, is present in up to 7% of pediatric tumors, compared with a lower reported range of 0–3% in adults [73]. Half of the cases (3/6) evaluated by Franco were of the follicular subtype, the remaining being classical PTC (n = 2) and the diffuse sclerosing PTC (n = 1) [82]. Other studies have also shown the association of these fusions with the follicular subtype [1,77]. A solid growth pattern in PTC has also been associated with *NTRK* fusions [85].

Only a handful of studies have explored the role of *PIK3CA* mutations in sporadic PTCs. In one study, these were found in 2 of the 79 PTC cases assessed by direct sequencing [72]. In another, *PIK3CA* mutations co-existed with *BRAF* V600E or *NRAS* Q61R, respectively, in two PTC cases [72,82]. Similarly, there are limited data on the status of *DICER1* mutations in pediatric PTCs, with a single study from Korea reporting a frequency of 7.6% in their PTC cohort [83]. 

An age-dependent variation exists for the molecular profile among pediatric thyroid carcinomas. Lee et al. [83] from Republic of Korea comprehensively characterized age-associated genetic alterations in a large cohort of pediatric PTCs. They divided their pediatric patients into three age groups (<10 years, 10–15 years and 15–20 years). Fusions occurred at a frequency of 92.9%, 27.5%, and 13.5%, respectively, in the different age groups. The frequency of *RET* fusions decreased with increasing age. Point mutations (*BRAF* V600E, *TERT*, *DICER1* and *RAS*) were observed in 7.1%, 30.0%, and 67.3%, respectively. Of these, *BRAF* V600E mutation was the most frequent, seen in 0%, 27.5%, and 57.7%, respectively. Notably, none of their cases showed *RAS* mutations [83].

Research on the role of molecular alterations as prognostic biomarkers in pediatric PTC is still in its infancy. A recent study investigated predictors of cervical lymph node metastases. While 68% of patients requiring neck dissection had somatic mutations, only 38% of those without lymph node metastases revealed molecular alterations. The difference was significant on univariate statistical analysis. The authors, hence, suggested that genetic mutation status is a predictor of nodal spread, and such patients should be kept on close follow-up if neck dissection was not initially required [86]. 

### 4.2. Follicular Thyroid Carcinoma

FTC is rare in the pediatric age group [87]. It presents with a larger mean tumor size, but with a favorable clinical outcome in contrast to adults [88,89]. In a study from Japan, Ito followed up 292 minimally invasive and 79 widely invasive FTC patients for a mean duration of 127 (6–339) and 123 months (3–332 months), respectively. Patients younger than 20 years were less likely to die of disease, irrespective of recurrence status [89]. 

Studies on FTC involving adult patients demonstrated RAS mutations (10–57%) and PAX8/PPARG fusions (up to 35–50%) to be the key players [90,91,92,93,94]. There are limited data on the molecular profile of pediatric FTC [14]. Vuong from Japan investigated a substantial cohort of 41 patients aged < 21 years. NRAS mutations were present in 12% and PAX8/PPARG fusions in none [88]. Studies from the West have also found a lower prevalence of 20–22% for RAS mutations and 0–20% for PAX8/PPARG fusions. However, the studies had a relatively small sample size [67,95]. 

Franco used ThyGeNEXT to assess 6 FTC cases in patients < 18 years of age. *HRAS* G13R, *HRAS* Q61R, and *KRAS* G12V were detected in one case each (3/6; 50%). Of their 47 benign non-neoplastic and neoplastic lesions, five showed molecular alterations. *GNAS* mutations were found in a case of multinodular goiter and two cases of follicular adenoma. One of the latter two cases showed an additional *PAX8/PPARG* translocation. A third follicular adenoma harbored *PTEN* mutation, and *TERT* C288T was identified in a case of diffuse hyperplasia. Notably, RAS mutations, often detected in adult benign nodules, were absent among their cases [82]. Using NGS, Ballester found a *CTNNB1* (β-catenin) p.S45P mutation in the single case of FTC assessed by them [96]. Among thyroid tumors, *CTNNB1* mutations have been reported primarily in PTC with fibromatosis/fasciitis-like/desmoid-type stroma [97], and ATC as a late event involved in cancer progression [98]. There are limited data on the role of β-catenin in FTC. Cell culture studies have revealed β-catenin activation to be dependent on PI3K/AKT activity, a pathway involved in FTC [99]. Another molecule of the Wnt/β-catenin signaling pathway which has been evaluated in FTC is Wnt-5a, an activator of the non-canonical Wnt pathways. When compared with normal thyroid tissue, experimental studies have revealed overexpression of Wnt-5a in FTC. The molecule promotes mesenchymal–epithelial transition by inducing cadherin expression and re-localization of β-catenin from the nuclei to the membrane [98,100].

FTC has also been associated with mutations in phosphatase and tensin homolog deleted on chromosome ten (*PTEN*), a tumor suppressor gene located at chromosome 10q23.3. Heterozygous germline mutation of *PTEN* leads to PTEN hamartoma tumor syndrome, an autosomal dominant disorder. There is a predisposition to developing malignancies in various organ systems [43]. FTC occurs in about 25% of carriers of *PTEN* mutation [101] and is one of the major criteria for the diagnosis of PTEN hamartoma tumor syndrome [102]. It has been recommended that all children diagnosed with FTC should undergo genetic counselling and testing for germline *PTEN* mutation [6]. Alzahrani and colleagues are the only ones to have studied *PTEN* in sporadic pediatric PTC patients and found exon 5 (c.295G > A) mutation in a single patient (1.4%) [72].

*DICER1* is another gene which has recently been implicated in the pathogenesis of pediatric FTC. The reported frequency has ranged from 25–53% [47,49]. Importantly, *DICER1* alterations are associated with the macrofollicular subtype of FTC [50]; hence, there is a need to evaluate young patients with this FTC variant for *DICER1* alterations.

Table 1 summarizes the differences in the clinical, pathological and molecular characteristics of adult and pediatric DTCs, and Table 2 details the literature available from Asia on the molecular alterations found in pediatric DTC. 

### 4.3. Poorly Differentiated Thyroid Carcinoma

There are negligible data on pediatric PDTC [7,51,68,81]. Mitsutake did not find any of the assessed driver mutations, namely *BRAF* (exon 15), H/*K*/*NRAS* (codons 12, 13 and 61), *TERT* promoter (C250T and C228T), *RET*/*PTC1*, *RET*/*PTC3*, *AKAP9*/*BRAF* or *ETV6* (exons 4 and 5)/*NTRK3* rearrangements in the single case of PDTC evaluated by them as a part of their larger cohort containing tumors detected following the accident at the Fukushima Daiichi Nuclear Power Plant in Japan [68]. In another study from Japan, 31 Japanese and 48 post-Chernobyl Ukrainian thyroid carcinomas involving children, adolescents and young adults were evaluated for *BRAF* V600E and *RAS* mutations. The single case of PDTC included was found to harbor *BRAF* V600E mutation. This tumor was focally immunopositive for CD15 and suggestive of dedifferentiation from PTC [81]. In a study from Saudi Arabia, Sanger sequencing did not reveal *BRAF* V600E and *TERT* promoter mutations in the single case of pediatric PDTC evaluated as a part of a mixed cohort of pediatric thyroid cancers [7]. Interestingly, instead of these known driver mutations, a high prevalence (83%; 5/6) of *DICER1* mutations was documented by Chernock. Additional mutations were found in *ATM*, *CDC73*, *TP53*, *MAP2K2*, *RBM10*, *ARID1A*, *FLT3*, and *EGFR* genes. None of the cases had *BRAF*, *RAS*, *TERT*, or *RET*/*PTC* alterations [51].

### 4.4. Medullary Thyroid Carcinoma

MTC is rare in children, having an annual incidence of 0.03 per 100,000 [57]. In contrast to adults, pediatric patients are more likely to have localized disease (70% vs. 52%), negative regional lymph nodes (48% vs. 31%), and a better 10-year cancer-specific survival rate (80% vs. 96%) [115]. While most (65–75%) of the cases in adults are sporadic, pediatric cases usually occur as a part of autosomal dominant syndromes associated with gain-of-function germline mutations in the *RET* proto-oncogene [18,57]. MEN type 2A syndrome (Sipple’s syndrome) is the most frequent. It is highly penetrant and usually presents before six years of age. The mutations involve the extracellular cysteine-rich region of the RET tyrosine kinase receptor, usually in exon 10 (codons 609, 611, 618 or 620) or exon 11 (codon 634). Bilateral pheochromocytomas and hyperparathyroidism are other common features of this syndrome [57,116]. Patients with MEN2B syndrome are also predisposed to developing MTC and pheochromocytoma. They may also develop gastrointestinal ganglioneuromas, oral and conjunctival mucosal neuromas and a marfanoid habitus. The mutation, usually Met918Thr in exon 16, occurs in the tyrosine kinase domain in the intracellular portion of the receptor, leading to ligand-independent catalytic activity. The mutation can be either inherited (25%) or arise de novo (75%) [59,117,118]. Patients with MEN2B develop MTC very early, within the first year of life, and have an average life expectancy of about 21 years [116]. Hence, prophylactic thyroidectomy is recommended in the first year of life [57,118]. 

Familial MTC (FMTC) harbors mutations similar to MEN2A, involving either the extracellular or the intracellular domain of the tyrosine kinase receptor; it is now considered an MEN2A variant. As it has less clinical penetrance; MTC is usually the sole clinical presentation. The tumor is also less aggressive and manifests in the second or third decade of life [57]. 

When familial, MTC is associated with a precursor lesion, the C-cell hyperplasia. The tumors are multifocal, bilateral and typically located at the junction of the upper one-third and the lower two-thirds of the thyroid lobes. As the risk of development and progression of MTC are related to the mutated codon, the management protocol of these patients is decided based on the variant present. MEN2B patients, having a mutation in the RET codon M918T, have the highest risk of developing MTC, and should be subjected to prophylactic thyroidectomy within the first few months up to the first year of life. The high-risk category includes patients with mutations in A883F or C634 codons. They should undergo thyroidectomy by 5 years of age; the timing and extent of the surgery are guided by serum calcitonin levels. The rest of the mutations have a moderate risk of disease. Children in the moderate risk category can undergo thyroidectomy either when serum calcitonin levels rise, or earlier if parents desire [118]. 

## 5. Role of miRNA in Pediatric Thyroid Carcinoma

MicroRNAs (miRNA) are small non-coding RNAs that regulate post-transcriptional gene expression. The utility of miRNA as potential diagnostic, prognostic, and therapeutic biomarkers in adult thyroid malignancies has been widely explored [43]. However, the literature on pediatric patients is limited to a single study [82]. The thyroid miRNA classifier (ThyraMIR) (Interpace Diagnostics, Parsippany, NJ, USA) is a proprietary commercially available molecular diagnostic test that uses relative expression of 10-miRNA (miR-222-3p, miR-146b-5p, miR-375, miR-29b-1-5p, miR-31-5p, miR-138-1-3p, miR-139-5p, miR-155, miR-204-5p, and miR-551b-3p) to stratify thyroid nodules categorized indeterminate on cytology [119]. Franco and colleagues used the classifier to assess 113 tumors, including 47 benign and 66 primary thyroid carcinomas, from patients < 18 years of age. The same tumors were also subjected to ThyGeNEXT. The ThyraMIR algorithm could correctly identify 59% of the malignant lesions. None of the benign lesions had a positive ThyraMIR test result. There were five miRNAs: miR-31-5p, miR-146b-5p, miR-222-3p, miR-375, and miR-551b-3p, which were significantly overexpressed in malignant nodules when compared with benign lesions. A positive result was also significantly associated with intrathyroidal spread, extrathyroidal extension and lymph node metastasis. MicroRNA expression patterns also varied with the histomorphological type of the cancers, and were similar to those reported in adults. Furthermore, 11 of the 39 cases characterized as malignant by ThyraMIR were found to be negative for mutations by ThyGeNEXT. The sensitivity of ThyGeNEXT increased from 53% to 70% when used in combination with ThyraMIR. Most malignant tumors found negative by molecular testing lacked aggressive histopathological features. However, the authors suggested the need for optimizing the ThyraMIR classifier for pediatric thyroid nodules [82]. The results need further validation to determine the accuracy and application of miRNA for diagnostic, therapeutic and prognostic purposes in pediatric patients.

## 6. Other Potential Biomolecules

PTCs, especially when BRAF-mutated, usually have a diminished expression of the thyroid differentiation genes SLC5A5 (sodium/iodide symporter, or NIS), apical iodide transporter, thyroperoxidase and thyroglobulin. In contrast, GLUT-1 expression increases. This phenotype interferes with RAI responsiveness at both diagnostic and therapeutic levels, wherein they are visible on 18-fluorodeoxyglucose positron emission tomography imaging but are not RAI avid [120]. Pediatric thyroid carcinomas are more often RAI responsive, possibly due to the increased expression of NIS in this group of patients [17]. However, a recent study documented variation in the expression of genes regulating thyroid hormone synthesis, namely SLC5A5, pendrin (PDS) and thyroid-stimulating hormone receptor (TSHR), across different age groups. They found the expression of NIS in DTC involving children < 10 years to be similar to adult tumors (>18 years). The expression levels of PDS and TSHR were lower in DTCs from children compared with adolescents (10–18 years) and adults [121]. Interestingly, a study from Korea found pediatric fusion-positive PTCs in children < 10 years of age to have a lower expression of thyroid differentiation genes, including SLC5A5, than adult fusion-positive PTCs. Two of their patients with fusion-positive and RAI-refractory tumors had low NIS expression. The patients responded well to fusion-targeted therapy with a decrease in tumor size and restoration of RAI uptake [83]. 

The other immunohistochemical biomarkers that have been evaluated in thyroid pathology, primarily to aid the differential diagnosis of PTC from its benign mimics, include galectin-3, cytokeratin 19 and Hector Battifora mesothelial-1 (HBME-1). Galectin-3, owing to its postulated role in the pathogenesis of PTC, is commonly used as an ancillary immunohistochemical marker to aid PTC diagnosis. HBME-1, originally used to confirm a mesothelial origin, has also found utility in PTC diagnosis. However, use of it in conjunction with other antibodies, instead of as a stand-alone marker, is preferred. When used individually, the reported sensitivities and specificities of HBME-1, galectin-3 and CK 19 for diagnosing thyroid malignancies have varied between 75 to 100% and 70 to 85%, respectively. When the three are used in conjunction, there is an increase in the reported diagnostic accuracy [122,123,124]. Owing to the availability of advanced techniques, several other new molecules are gaining recognition as potential biomarkers in thyroid oncology practice [43]. However, most studies are adult-centric. There are, as of now, not enough prospective and multi-center studies to allow strong recommendations regarding their role in pediatric patients. 

## 7. Targeted Therapy in Pediatric Patients

While treating pediatric thyroid cancers, it should be kept in mind that their unique molecular makeup precludes the applicability of adult-standardized guidelines. Hence, molecular profiling is useful in cases where targeted therapy is being considered. These include metastatic symptomatic cancers that cannot be controlled with localized therapy, or progressive cases of RAI-refractory DTC. 

The common kinase inhibitors used in the treatment of thyroid malignancies include Sorafenib (BRAF inhibitor), Lenvatinib (anti-VEGFR, anti-FGFR, anti-PDGFR and RET inhibitor), Cabozantinib (anti-VEGFR), Donafenib (multikinase inhibitor), and Vandetanib (RET and VEGFR2 and 3 inhibitor). As these are associated with significant side effects, drugs targeting more specific mutations, such as gene fusions, are fast evolving. The TRK inhibitors Entrectinib and Larotrectinib have been recommended by the FDA for use in both pediatric and adult tumors with *NTRK* fusions. Larotrectinib is selective for TrkA, TrkB and TrkC, whereas Entrectinib also inhibits ROS1 and ALK [125,126]. Selpercatinib has shown efficacy in *RET* fusion-positive pediatric PTCs [83]. Restoration of radioiodine uptake by these drugs suggests the occurrence of redifferentiation in these tumors. However, the acquisition of drug resistance is a major obstacle. To overcome this, trials of novel agents such as Selitrectinib (LOXO-195), Repotrectinib (TPX-0 0 05) and Taletrectinib (DS-6051b) are currently underway [126]. 

## 8. Molecular Evaluation of Indeterminate Thyroid Nodules in Pediatric Patients

Knowledge of the molecular landscape of pediatric thyroid nodules is still evolving. While a positive mutational test would be more likely to be associated with malignancy, a negative genetic test does not reliably exclude it. In contrast to adult cases, owing to a lack of validation, the ATA does not recommend molecular testing on cytology material in pediatric thyroid nodules. Additionally, the reported risk of malignancy (ROM) in nodules with indeterminate cytology is higher in children (up to 30%) than in adults (5–15%). Hence, the ATA recommends upfront surgery (lobectomy with isthmusectomy) in these patients [6,69]. Wang et al. have challenged this approach. Their meta-analysis of pediatric thyroid nodules revealed that the two indeterminate Bethesda categories (atypia of undetermined significance/follicular lesion of undetermined significance and follicular neoplasm/suspicious for a follicular neoplasm) had lower malignancy rates than previously reported. Moreover, there is a higher risk of post-thyroidectomy complications in children. Thus, the authors asserted the need for management guidelines specific to the Bethesda category, instead of upfront surgery [127].

The antibodies galectin-3, HBME-1 and CK19 have also shown promise in the better characterization of the cytologically indeterminate thyroid nodules [123,128]. Although these antibodies have not been exclusively studied in pediatric patients, studies recommending this cocktail have pediatric samples in their study cohort [129]. Other researchers have advised molecular testing to help in surgical decision-making [12,95,130]. There is limited evidence to support the utility of miRNA-based classification in pediatric thyroid cytology. Franco and co-authors demonstrated a specificity of 100% for miRNA testing to detect malignancy. Therefore, they suggested using this assay in nodules with indeterminate cytology to increase diagnostic accuracy and improve risk stratification [82]. 

## 9. Conclusions

Despite being the largest contributor to new cases of pediatric thyroid carcinomas, there are limited molecular studies from Asia; the majority of the published literature is from the West.

Instead of *BRAF* and *RAS* point mutations common in adult cases, irrespective of the radiation status, chromosomal rearrangements are more frequent in pediatric PTCs. *RET/PTC* gene rearrangement is the most common, followed by *BRAF* fusions. *TERT* promoter mutations, markers of aggressive disease in adults, are uncommon in children. There is a dearth of data on FTC, PDTC and ATC. Preliminarily, *DICER1* mutations appear to be the key players in pediatric FTCs and PDTC. MTC in children needs evaluation to rule out a syndromic association. Non-MTC follicular cell-derived tumors can also be rarely familial. The literature on the role of microRNA as a biomarker in pediatric thyroid carcinomas is scarce. Any influence of the molecular profile on the expression of thyroid differentiation genes also remains unconfirmed. 

Due to the rarity of pediatric thyroid carcinomas, their molecular landscape is still incompletely decoded. The cost and availability of high throughput techniques, especially in the developing world, are other roadblocks. Understanding the applicability of their molecular characteristics for diagnosis, prognostication, and therapeutics requires multi-institutional studies utilizing sensitive and high-performance molecular techniques. 

## Figures and Tables

**Figure 1 diagnostics-12-03136-f001:**
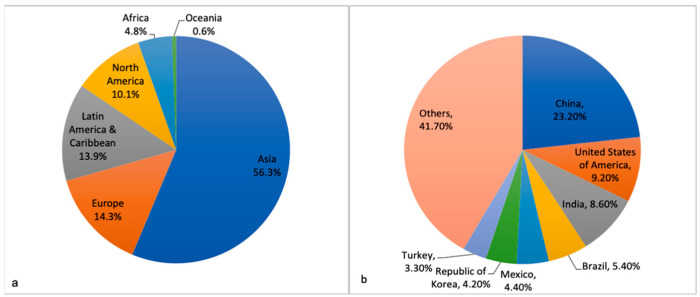
(**a**) Continent-wise and (**b**) Country-wise, percentage of new cases of thyroid cancer between 0–19 years; adapted from GLOBOCAN 2020 [20].

**Table 1 diagnostics-12-03136-t001:** Clinical, pathological and molecular characteristics of adult and pediatric thyroid carcinomas.

Characteristics	Adults	References	Pediatric Age Group	References
**Clinical symptoms**	Hoarseness, dysphagia, cough. Nodule common but rarely (<5–10%) malignant	[14,59]	Asymptomatic, incidentally detected. Nodule uncommon but frequently (22–26%) malignant if present	[14,59]
**Upfront lymph node involvement**	20–50%	[103]	40–90%	[103]
**Distant metastasis**	2%	[103]	20–30%	[103]
**Radiation exposure**	Risk of cancer less	[104]	5–10 fold increased risk; higher rates reported post-radiotherapy	[104,105]
**FNAC** USG guidance Nodule size Hyperfunctioning nodule The Bethesda system for reporting ROM Bethesda III Recommended management Bethesda III ROM Bethesda IV Recommended management Bethesda IV Molecular testing of nodules with indeterminate cytology	Not mandatory <1cm: FNAC usually not recommended FNAC not recommended Recommended 6–18% Repeat FNAC/molecular testing/lobectomy 10–40% Molecular testing/lobectomy Recommended	[6] [6,103] [6] [6,15] [14,106] [106] [14,106] [106] [69]	Mandatory Size not a criterion for decision making FNAC not recommended (should be removed surgically) Recommended 28% Lobectomy plus isthmusectomy 58% Lobectomy plus isthmusectomy Not recommended	[6] [6,103] [6] [6,15] [6,14,15] [6] [14] [6] [6,69]
**Tumor classification system**	AJCC TNM classification	[6]	AJCC TNM classification with ATA risk-stratification system *	[6]
**5-year relative survival**	98.3%	[3]	99.7%	[3]
**PTC** **multifocality**	20%	[107]	65%	[6]
**Histopathological subtypes of PTC**	High risk subtypes less common (<20%)	[108]	Classic PTC 20–50%; high risk subtypes (tall cell, diffuse sclerosing, solid/trabecular) form 15–40%	[14,60,61,109]
**Molecular profile (PTC)** *BRAF* V600E	30–90%	[107]	0–63% (sporadic) 0–70% (post-radiation)	[14,73]
*BRAF* fusions	<3%	[63,73]	0–20% (sporadic) 0–11% (post-radiation)	[14,73]
*RET* fusions	5–35%	[107]	22–65% (sporadic) 33–77% (post-radiation)	[62,73]
*H-/K-/N-RAS* mutations	0–35%	[107]	<10%	[1,72]
*TERT* promoter mutations (C250T, C228T)	5–25%	[107]	0–27%	[73,83]
*NTRK* fusions	1–5%	[73]	0–20% (sporadic) 1–15% (post-radiation)	[73,77]
*PAX8/PPARG* fusion	0–5%	[73]	0–9% (sporadic) 4% (post-radiation)	[73]
*DICER1* mutations	∼4% **	[110]	7.6%	[83]
*ALK* fusions	0–3%	[73]	0–7% (sporadic) 1–7% (post-radiation)	[73]
**Molecular profile (FTC)** *H-/K-/N-RAS* mutations	10–57%	[90,91,93,94]	0–50%	[14,82,88]
*PAX8/PPARG* fusion	35–50%	[90,92,93,94]	0–20%	[67,88,95]
*PTEN* mutations	<1%	[63]	<2% (sporadic) 25% (carriers of *PTEN* mutation)	[72,101]
*DICER1* mutations	1%	[47]	25–53% ***	[47,49]

FNAC, fine needle aspiration cytology; USG, ultrasonography; ROM, risk of malignancy; AJCC, American Joint Committee on Cancer; TNM, tumor node metastasis; ATA, American Thyroid Association; PTC, papillary thyroid carcinoma; FTC, follicular thyroid carcinoma. * For PTC. ** 85.7% germline. *** 50% germline in one study [49].

**Table 2 diagnostics-12-03136-t002:** Studies from Asian countries with details of genetic alterations in pediatric patients.

Studies	Year	Country	Age Range (Years)	n	Sporadic/Post Radiation	Subtypes	Molecular Method Used	*BRAF* V600E	*RET*/*PTC*	*BRAF* Fusion	*TERT* Promoter Mutation	*RAS* Mutation	*NTRK* Fusions	Additional Alterations
Motomura et al. [111]	1998	Japan	9–14	10	Sporadic	Classical (6) F-PTC (2) DS-PTC (1) S-PTC (1)	RT-PCR followed by southern hybridization	-	*RET/PTC1* 20% *RET/PTC3* 10%	-	-	-	-	-
Kumagai et al. [81]	2005	Japan	<15	31	Sporadic	Classical (27) F-PTC (2) FTC (1) PDTC (1)	Direct sequencing, RT-PCR	3.2%	-	-	-	0	-	-
			≤15	15	Radiation ^#^	Classical (2) S-PTC (4) F-PTC (2) Mixed (7)		0	-	-	-	0	33.3%	-
			15–31	33	Radiation ^#^	Classical (7) S-PTC (1) F-PTC (9) Mixed (16)		24.2%	-	-	-	6.1%	36.4%	-
Mitsutake et al. [68]	2015	Japan	≤22	68	Likely sporadic ^$^	Classical (61) F-PTC (2) CMTC (4) PDTC (1)	Direct sequencing, RT-PCR	63.2%	*RET/PTC1* 8.8% *RET/PTC3* 1.5%	0	0	0	5.9%	-
Alzahrani et al. [7]	2016	Saudi Arabia	≤18	55	Sporadic	Classical (44) FV (6) TC-PTC (1) DS-PTC (1) FTC (2) PDTC (1)	Direct sequencing	22.6%	-	-	1.8%	-	-	-
Alzahrani et al. * [72]	2017	Saudi Arabia	≤18 yrs	79	Sporadic	Classical (72) F-PTC (7)	Direct sequencing	24.1%	-	-	1.3%	2.5%	-	*PIK3CA* exon 9: 1.4% *PIK3CA* exon 20: 1.3% *PTEN* exon 5: 1.4%
Geng et al. ° [112]	2017	China	3–13	48	Sporadic	Classical (41) F-PTC (5) DS-PTC (2)	Direct sequencing	35.4%	-	-	-	-	-	-
Oishi et al. [65]	2017	Japan	≤20	81	Sporadic	Classical (66) CMTC (1) F-PTC (2) DS-PTC (4) S-PTC (8)	Allele specific PCR and/or Sanger sequencing	54%	-	-	0%	-	-	-
Vuong et al. [88]	2017	Japan	<21	41	Sporadic	FTC (41)	Direct sequencing and RT-PCR	-	-	-	-	12.2%	-	0% (*PAX8/PPARG*)
Geng et al. ° [84]	2019	China	3–13	48	Sporadic	Classical (41) F-PTC (5) DS-PTC (2)	Direct sequencing	-	-	-	27.1%	-	-	-
Kure et al. [113]	2019	Japan	13–19	7	Sporadic	Classical (5) CMTC (1) Classical with poorly differentiated component (1)	Direct sequencing	29%	-	-	-	-	-	-
Iwadate el al. ^&^ [114]	2020	Japan	0–24	138	Likely sporadic ^$^	Classical (125) CMTC (4) F-PTC (3) DS-PTC (2) S-PTC (2) PDTC (1) Others (1)	Direct sequencing and RT-PCR	70.6% of PTCs	*RET/PTC1* 5.9% of PTCs *RET/PTC3* 0.7% of PTCs	-	-	-	5.9% of PTCs	*AFAP1L2/RET* (0.7% of PTCs) *PPFIBP2/RET* (0.7% of PTCs) *STRN/ALK* (1.5% of PTCs) *KIAA1217/RET* (0.7% of PTCs) *Delta RFP/RET* (0.7% of PTCs)
Chakraborty et al. [71]	2020	India	≤20	100	Sporadic	Classical (72) F-PTC (24) TC-PTC (2) FTC (2)	Direct sequencing	14%	-	-	0%	-	-	-
Lee et al. [49]	2020	Republic of Korea	<20	15	10 sporadic, 4 DICER1 syndrome, 1 PTEN hamartoma syndrome	FTC	WES, targeted NGS, direct sequencing	-	-	-	0%	0%	-	DICER1 (53.3%), PTEN (6.7%), PAX8/PPARG (6.7%)
Bae et al. [47]	2021	Japan, Republic of Korea	<18	41	Sporadic	Follicular-patterned tumors ∗	Targeted NGS	0%	0%	0%	0%	20%	0%	DICER1 (22%), FGFR3 (15%), PTEN (12%), STK11 (10%), APC (5%), TSHR (5%), CTNNB1 (2%), TP53 (2%), EIF1AX (2%), FGFR4 (2%), GNAS (2%), RET (2%), SOS1 (2%), THADA/IGF2BP3 (2%)
Lee et al. [83]	2021	Republic of Korea	<10	14	Sporadic and radiation ^	Classical (75) DS-PTC (14) Others (15)	NGS, direct sequencing, FISH, and/or IHC.	0%	64.3%	-	7.1%	0%	14.3%	0% (*DICER1*)
			10–15	40				27.5%	20%	-	0%	0%	5%	2.5% (*DICER1*)
			15–20	52				57.7%	7.7%	-	1.9%	0%	0%	7.7% (*DICER1*)

Abbreviations: Classical: classical PTC; CMTC: cribriform morular thyroid carcinoma; DS-PTC: diffuse sclerosing subtype of PTC; FISH: fluorescence in situ hybridization; FTC: follicular thyroid carcinoma; F-PTC: follicular subtype of PTC; Hobnail: hobnail subtype of PTC IHC: immunohistochemistry; NGS: next generation sequencing; PDTC: poorly differentiated thyroid carcinoma; S-PTC: solid subtype of PTC; TC-PTC: tall cell subtype of PTC ^#^ post-Chernobyl, ^$^ post-Fukushima Daiichi Nuclear Power Plant accident, but unlikely to be radiation-induced. * Includes patients from their previous study [7]. ° Includes the same cohort in both studies. ^&^ Includes patients from their previous study [68]. ^ Prior history of radiotherapy.

## Data Availability

Not applicable.

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
