# Peer review of "Molecular Landscape of Pediatric Thyroid Cancer: A Review"

_diagnostics, 2022, doi:10.3390/diagnostics12123136_

Round 1

Reviewer 1 Report (New Reviewer)

I think this review reports an interesting observation about molecular landscape of paediatric thyroid cancer. The narrative of the manuscript is well presented and easy to follow.

Manuscript focused on epidemiology, risk factors and molecular profile of peadiatric thyroid cancer. Given the extensive discussion of current knowledge regarding the molecular basis of thyroid cancers in children, it is necessary to supplement these data with information on the availability of molecularly targeted therapies and selective inhibitors. 

Special attention should be paid to the possibility of RET and NTRK inhibitor-based therapies. The section on targeted therapy is inadequately described in this manuscript.

Author Response

Point 1: Manuscript focused on epidemiology, risk factors and molecular profile of paediatric thyroid cancer. Given the extensive discussion of current knowledge regarding the molecular basis of thyroid cancers in children, it is necessary to supplement these data with information on the availability of molecularly targeted therapies and selective inhibitors. 

Special attention should be paid to the possibility of RET and NTRK inhibitor-based therapies. The section on targeted therapy is inadequately described in this manuscript.

Response 1: We thank the reviewer for the valuable comments. As advised, we have elaborated on the RET and NTRK inhibitor-based therapies in the section on targeted therapies.

Reviewer 2 Report (New Reviewer)

This is a well written and interesting review on thyroid tumors affecting pediatric patients.

major points:

The study is centered mostly on the molecular genetic rather than the more generic molecular landscape of pediatric thyroid tumors. The authors, in fact, have exposed with extensive details the genetic biomarkers associated with all the different types of thyroid malignancy in pediatric patients, including point mutations, fusions, miRNA, and thyroid differentiation genes. They have included other biomolecules potentially involved in a dedicate short paragraph in which they have mentioned molecules potentially involved in malignant transformation and therefore, useful as diagnostic markers. Among these they have mentioned the SLC5A5 (sodium/iodide symporter, or NIS), the apical iodide transporter, the thyroperoxidase, the thyroglobulin, the GLUT-1, the Pendrin (PDS) and the Thyroid Stimulating Hormone Receptor (TSHR). However, there is no mention regarding the use of other molecules that proved to be accurate in thyroid cancer in adult patients, such as Galectin-3 (see Ref. Lancet Oncology 2008) and HBME-1 (see Ref. de Matos et al. Diagnostic Pathology 2012, 7:97) as well as CD133, cytokeratin-19, CD44v6, HMGI(Y) and telomerase. Although there are no enough prospective and multicenter high-quality studies to allow strong recommendations regarding the role of these markers in thyroid cancer in pediatric patients, the documented role of these molecules in thyroid cancer diagnosis in adult patients and the fact that current management guidelines for pediatric patients are largely derived from those for adults, as stated by the authors, indicate that they deserve a mention in this review. In addition, some of them, and in particular the Galectin-3, are useful in the diagnostic evaluation of cytologically indeterminate thyroid nodules, that occur in a percentage ranging from 13% to 43% in pediatric patients (See Ref. Cancer Cytopathol 2019;127:231-239). The authors should include these protein markers in the paragraph entitled: “other potential biomolecules” and report those studies that have analyzed them in pediatric patients.

 The paragraph entitled “Molecular Evaluation of Indeterminate Thyroid Nodules in Paediatric Patients” is a rather short one, with few information reported. Only a brief mention was reported regarding the utility of miRNA-based classification of such nodules. Again, no indication was given regarding other protein markers that are already in clinical practice for the management of such nodules.

It is not clear why a repeat FNA should be more cumbersome compared to surgery (lobectomy with isthmusectomy) in children. The authors should give a clear explanation of this sentence or remove it.

Author Response

Point 1: There is no mention regarding the use of other molecules that proved to be accurate in thyroid cancer in adult patients, such as Galectin-3 (see Ref. Lancet Oncology 2008) and HBME-1 (see Ref. de Matos et al. Diagnostic Pathology 2012, 7:97) as well as CD133, cytokeratin-19, CD44v6, HMGI(Y) and telomerase. Although there are no enough prospective and multicenter high-quality studies to allow strong recommendations regarding the role of these markers in thyroid cancer in pediatric patients, the documented role of these molecules in thyroid cancer diagnosis in adult patients and the fact that current management guidelines for pediatric patients are largely derived from those for adults, as stated by the authors, indicate that they deserve a mention in this review.

Response 1: We thank the reviewer for the very valid inputs. As advised, we have included paragraph on the utility of protein markers in the diagnosis of thyroid malignancies. The importance of telomerase in thyroid malignancy had already been discussed in the manuscript under TERT promoter mutations. As appropriately advised by the reviewer, we have also added a statement acknowledging that there are no enough prospective and multicenter high-quality studies to allow strong recommendations regarding the role of these markers in thyroid cancer in pediatric patients.

Point 2: In addition, some of them, and in particular the Galectin-3, are useful in the diagnostic evaluation of cytologically indeterminate thyroid nodules, that occur in a percentage ranging from 13% to 43% in pediatric patients (See Ref. Cancer Cytopathol 2019;127:231-239). The authors should include these protein markers in the paragraph entitled: “other potential biomolecules” and report those studies that have analyzed them in pediatric patients.

Response 2: A paragraph on the utility of the above-mentioned antibodies in the diagnosis of indeterminate thyroid nodules has been included in the subsection, as advised.

Point 3: The paragraph entitled “Molecular Evaluation of Indeterminate Thyroid Nodules in Paediatric Patients” is a rather short one, with few information reported. Only a brief mention was reported regarding the utility of miRNA-based classification of such nodules. Again, no indication was given regarding other protein markers that are already in clinical practice for the management of such nodules.

Response 3: As advised, the paragraph on molecular evaluation of indeterminate thyroid nodules has been edited with inclusion of all the points mentioned.

Point 4: It is not clear why a repeat FNA should be more cumbersome compared to surgery (lobectomy with isthmusectomy) in children. The authors should give a clear explanation of this sentence or remove it.

Response 4: The point was well taken and as advised, we have deleted the lines from the manuscript, and edited the paragraph appropriately.

Round 2

Reviewer 1 Report (New Reviewer)

Thank you for considering my suggestions.

Reviewer 2 Report (New Reviewer)

The authors have satisfactorily addressed most of my concerns. In particular, they have added a mention regarding the proteins potentially involved in malignant transformation and used as diagnostic markers. 

This manuscript is a resubmission of an earlier submission. The following is a list of the peer review reports and author responses from that submission.

Round 1

Reviewer 1 Report

This is a nice review on pediatric thyroid cancer, which is quite comprehensive in its description of molecular aberrancies in these tumors, although not all papers in the field have been identified and cited.

Apart from a linguistic brush-up, the authors should consider the following remarks:

Line 76-77: The incidence of pediatric thyroid cancer is not 1.9 %, rather it is the fraction of all thyroid cancers being identified in children and adolescents.

Line 173: When discussing DICER1 mutations in thyroid neoplasms, it would be worth mentioning that these alterations are found to a high degree in the macrofollicular variant of FTC, not least in young patients. This is of importance as this finding should lead to considering looking at constitutional DICER1 mutations, esp in young patients.

Line 323-324: “ETV6/NTRK3 was noted in 11%”; preferable to write “in one single patient”, which is less misleading.

Line 368: “The only study with a substantial 368 cohort size is from Japan”; well, which size was it?

Line 445 and further: Most authors would advocate that the term FMTC should not be used today, as it turns out that families with RET mutations and (seemingly) isolated appearance of MTC indeed are MEN 2A families if you follow them long enough; plenty of families formerly classified as FMTC have transformed to MEN 2A when a pheo or pHPT suddenly appears in an individual.

Author Response

Response to Reviewer 1 Comments

Point 1:  This is a nice review on pediatric thyroid cancer, which is quite comprehensive in its description of molecular aberrancies in these tumors, although not all papers in the field have been identified and cited.

Response 1: We sincerely appreciate all valuable comments and suggestions. We have added more publications to the manuscript. These include:

  1. Bae, J.S.; Jung, S.H.; Hirokawa, M.; Bychkov, A.; Miyauchi, A.; Lee, S.; Chung, Y.J.; Jung, C.K. High Prevalence of DICER1 Mutations and Low Frequency of Gene Fusions in Pediatric Follicular-Patterned Tumors of the Thyroid. Endocr. Pathol. 2021, 32, 336–346, doi:10.1007/S12022-021-09688-9.
  2. Lee, Y.A.; Im, S.W.; Jung, K.C.; Chung, E.J.; Shin, C.H.; Kim, J. Il; Park, Y.J. Predominant DICER1 Pathogenic Variants in Pediatric Follicular Thyroid Carcinomas. Thyroid 2020, 30, 1120–1131, doi:10.1089/THY.2019.0233.
  3. Juhlin, C.C.; Stenman, A.; Zedenius, J. Macrofollicular variant follicular thyroid tumors are DICER1 mutated and exhibit distinct histological features. Histopathology 2021, 79, 661–666, doi:10.1111/HIS.14416.
  4. Chernock, R.D.; Rivera, B.; Borrelli, N.; Hill, D.A.; Fahiminiya, S.; Shah, T.; Chong, A.S.; Aqil, B.; Mehrad, M.; Giordano, T.J.; et al. Poorly differentiated thyroid carcinoma of childhood and adolescence: a distinct entity characterized by DICER1 mutations. Mod. Pathol. 2020, 33, 1264–1274, doi:10.1038/S41379-020-0458-7.
  5. Rooper, L.M.; Bynum, J.P.; Miller, K.P.; Lin, M.T.; Gagan, J.; Thompson, L.D.R.; Bishop, J.A. Recurrent DICER1 Hotspot Mutations in Malignant Thyroid Gland Teratomas: Molecular Characterization and Proposal for a Separate Classification. Am. J. Surg. Pathol. 2020, 44, doi:10.1097/PAS.0000000000001430.
  6. Nosé, V. DICER1 gene alterations in thyroid diseases. Cancer Cytopathol. 2020, 128, 688–689, doi:10.1002/CNCY.22327.
  7. Durieux, E.; Descotes, F.; Mauduit, C.; Decaussin, M.; Guyetant, S.; Devouassoux-Shisheboran, M. The co-occurrence of an ovarian Sertoli-Leydig cell tumor with a thyroid carcinoma is highly suggestive of a DICER1 syndrome. Virchows Arch. 2016, 468, 631–636, doi:10.1007/S00428-016-1922-0.
  8. Rivkees, S.A.; Mazzaferri, E.L.; Verburg, F.A.; Reiners, C.; Luster, M.; Breuer, C.K.; Dinauer, C.A.; Udelsman, R. The Treatment of Differentiated Thyroid Cancer in Children: Emphasis on Surgical Approach and Radioactive Iodine Therapy. Endocr. Rev. 2011, 32, 798, doi:10.1210/ER.2011-0011.
  9. Iglesias, M.L.; Schmidt, A.; Ghuzlan, A. Al; Lacroix, L.; de Vathaire, F.; Chevillard, S.; Schlumberger, M. Radiation exposure and thyroid cancer: a review. Arch. Endocrinol. Metab. 2017, 61, 180–187, doi:10.1590/2359-3997000000257.
  10. Livolsi, V.A. Papillary thyroid carcinoma: an update. Mod. Pathol. 2011 242 2011, 24, S1–S9, doi:10.1038/modpathol.2010.129.
  11. Bhatia, S.; Yasui, Y.; Robison, L.L.; Birch, J.M.; Bogue, M.K.; Diller, L.; DeLaat, C.; Fossati-Bellani, F.; Morgan, E.; Oberlin, O.; et al. High risk of subsequent neoplasms continues with extended follow-up of childhood Hodgkin’s disease: report from the Late Effects Study Group. J. Clin. Oncol. 2003, 21, 4386–4394, doi:10.1200/JCO.2003.11.059.
  12. Ali, S.Z.; Cibas, E.S. The Bethesda system for reporting thyroid cytopathology : definitions, criteria, and explanatory notes; 2nd Edition, 2018.
  13. Lloyd, R.V.; Osamura, R.Y.; Kloppel, G.; Rosai, J. Chapter 2 Tumours of the Thyroid Gland. In WHO Classification of Tumours of Endocrine Organ; International Agency for Research on Cancer (IARC): Lyon, France, 2017; pp. 65–143.108.
  14. Koo, J.S.; Hong, S.; Park, C.S. Diffuse Sclerosing Variant Is a Major Subtype of Papillary Thyroid Carcinoma in the Young. https://home.liebertpub.com/thy 2009, 19, 1225–1231, doi:10.1089/THY.2009.0073.
  15. Canberk, S.; Ferreira, J.C.; Pereira, L.; Batlsta, R.; Vieira, A.F.; Soares, P.; Sobrinho Simões, M.; Máximo, V. Analyzing the Role of DICER1 Germline Variations in Papillary Thyroid Carcinoma. Eur. Thyroid J. 2021, 9, 296–303, doi:10.1159/000509183.
  16. Zhao, Z.; Yin, X. D.; Zhang, X. H.; Li, Z.W.; Wang, D.W. Comparison of pediatric and adult medullary thyroid carcinoma based on SEER program. Sci. Rep. 2020, 10, 1–8, doi:10.1038/s41598-020-70439-7.

Point 2:  Apart from a linguistic brush-up, the authors should consider the following remarks:

Response 2: The entire manuscript has been edited using software and the language improvised upon.

Point 3:  Line 76-77: The incidence of pediatric thyroid cancer is not 1.9 %, rather it is the fraction of all cancers being identified in children and adolescents.

Response 3: We totally agree with the reviewer. The line has been edited as per the suggestion.

Point 4:  Line 173: When discussing DICER1 mutations in thyroid neoplasms, it would be worth mentioning that these alterations are found to a high degree in the macrofollicular variant of FTC, not least in young patients. This is of importance as this finding should lead to considering looking at constitutional DICER1 mutations, esp in young patients.

Response 4: We agree with the reviewer. The following sentences and relevant references have been added/edited:

In section 3.4 on ‘Familial/Genetic syndromes’:

DICER1 mutations have now been identified as important drivers in pediatric thyroid nodules, with an overall reported rate of 30%, in contrast to about 1% in adults [47]. These may be germline or somatic, and phenotypic presentation may be in the form of benign or malignant thyroid disease. The disease is usually multifocal and nodular, ranging from adenomatous goiter to true neoplasms. The latter include FA, PTC, FTC, PDTC, and the very rare carcinosarcoma, and thyroblastoma [48–53]. While macrofollicular architecture has been found to be associated with somatic alterations [50], co-occurrence of DTC and Sertoli-Leydig cell tumor has been suggested to be highly indicative of DICER1 syndrome [54].

In section 4.2 on ‘Follicular thyroid carcinoma’:

DICER1 is another gene, which has recently been implicated in the pathogenesis of pediatric FTC. The reported frequency has ranged from 25-53% [47,49]. Importantly, DICER1 alterations are associated with the macrofollicular variant of FTC [50]; hence, the need to evaluate young patients with this FTC variant for DICER1 alterations.

Point 5:  Line 323-324: “ETV6/NTRK3 was noted in 11%”; preferable to write “in one single patient”, which is less misleading.

Response 5: The line has been edited as appropriately suggested by the reviewer.

Point 6:  Line 368: “The only study with a substantial cohort size is from Japan”; well, which size was it?

Response 6: The statement has been refined to include the sample size.

Point 7:  Line 445 and further: Most authors would advocate that the term FMTC should not be used today, as it turns out that families with RET mutations and (seemingly) isolated appearance of MTC indeed are MEN 2A families if you follow them long enough; plenty of families formerly classified as FMTC have transformed to MEN 2A when a pheo or pHPT suddenly appears in an individual.

Response 7: We agree with the reviewer, and the paragraph has been edited, as suggested, to:

“Familial MTC (FMTC) harbors mutations similar to MEN2A, involving either the extracellular or the intracellular domain of the tyrosine kinase receptor, and is now considered a MEN2A variant. As it has less clinical penetrance, MTC is usually the sole clinical presentation. The tumor is also less aggressive, and manifests in second or third decade of life.”

Reviewer 2 Report

General comment.

This review deals with the molecular profiles of different types of thyroid cancer occurring in pediatric age. It also discusses how the cancer molecular profile can be used as novel biomarker in the clinical management of these patients.

The subject is relevant for both translational and clinical aspects and the presentation is complete and well organized. Cited references are well updated.

Specific comments

- line 81: “female preponderance” should be better specified because pre- and post-puberty cases may have a different F/M ratio.

- lines 86-87 and Fig.1 (continent wise and country wise distribution): “individual countries”. The sentence should be completed mentioning the incidence or the prevalence of pediatric thyroid cancers in the different countries. In the present form the influence of the size of the country population is determinant.

The authors should better discuss this point that is also affected by the different disease surveillance systems and by a variety of environmental factors.

Finally, the possible influence of the mixed ethnicity present in countries such as the US should also be mentioned.

The data regarding the observed frequency of genetic alteration are influenced by these factors, as evidenced in the following paragraphs of the review.

- lines 215-216: “these differences are related to significant differences in molecular genetics of pediatric….”. Difference in the genetic profile is certainly a factor but the different endocrine, metabolic and immune characteristics of the pediatric age must not be underestimated and should also be mentioned when discussing the differences in thyroid cancer incidence in different pediatric age categories.

- Table 2 is poorly presented and difficult to read. It should be reorganized.

- I do agree that the clinical use of molecular alterations as prognostic biomarkers in pediatric thyroid cancer (except for the medullary histotype) is still in its infancy (lines 348-349).

I suggest that this concept should be better underlined in paragraph 5 and 8, when discussing the use of molecular measurements as diagnostic (cytology), prognostic and therapeutic use. Cost and availability of these techniques, in addition to sensitivity and accuracy, should be mentioned. Today their use is limited to scientific programs.

- line 515: “adjusted for the age” (rather body weight?).

- lines 546-547: “ genotype of pediatric PTCs is essential for best management and targeted therapy”. Not yet the case

Author Response

Response to Reviewer 2 Comments

Point 1:  This review deals with the molecular profiles of different types of thyroid cancer occurring in pediatric age. It also discusses how the cancer molecular profile can be used as novel biomarker in the clinical management of these patients.

The subject is relevant for both translational and clinical aspects and the presentation is complete and well organized. Cited references are well updated.

Response 1: We are grateful to the reviewers for their inputs and appreciative words.

Point 2:  - line 81: “female preponderance” should be better specified because pre- and post-puberty cases may have a different F/M ratio.

Response 2: The point is well-taken, and the statement has been edited as follows:

“Similar to adults, a female preponderance is noted. Importantly, the incidence rates in males and females are 0.2 and 0.6 per 1,000,000 in the children aged 0-14 years, and 1.2 and 6 per 1,000,000 in the age group 15-19 years [3]. Hence, the differences in the incidence rates in males and females are more pronounced in the post-pubertal age group.”

Point 3:  - lines 86-87 and Fig.1 (continent wise and country wise distribution): “individual countries”. The sentence should be completed mentioning the incidence or the prevalence of pediatric thyroid cancers in the different countries. In the present form the influence of the size of the country population is determinant. The authors should better discuss this point that is also affected by the different disease surveillance systems and by a variety of environmental factors.

Response 3: We appreciate bringing this important point to our notice. The point has been discussed, as suggested.

Point 4:  Finally, the possible influence of the mixed ethnicity present in countries such as the US should also be mentioned.

Response 4: This point has also been discussed, as suggested.

Point 5:  The data regarding the observed frequency of genetic alteration are influenced by these factors, as evidenced in the following paragraphs of the review.

- lines 215-216: “these differences are related to significant differences in molecular genetics of pediatric….”. Difference in the genetic profile is certainly a factor but the different endocrine, metabolic and immune characteristics of the pediatric age must not be underestimated and should also be mentioned when discussing the differences in thyroid cancer incidence in different pediatric age categories.

Response 5: The point has been discussed, as suggested.

Point 6:  - Table 2 is poorly presented and difficult to read. It should be reorganized.

Response 6: The table has been reorganized and formatted.

Point 7:  - I do agree that the clinical use of molecular alterations as prognostic biomarkers in pediatric thyroid cancer (except for the medullary histotype) is still in its infancy (lines 348-349).

I suggest that this concept should be better underlined in paragraph 5 and 8, when discussing the use of molecular measurements as diagnostic (cytology), prognostic and therapeutic use. Cost and availability of these techniques, in addition to sensitivity and accuracy, should be mentioned. Today their use is limited to scientific programs.

Response 7: We sincerely acknowledge and appreciate the reviewer’s suggestion. The point made by the reviewer has been discussed in paragraphs 5, 7, 8 and also in the section ‘Conclusions’.

Point 8:  - line 515: “adjusted for the age” (rather body weight?).

Response 8: The statement has been corrected to “adjusted for body weight”.

Point 9:  - lines 546-547: “ genotype of pediatric PTCs is essential for best management and targeted therapy”. Not yet the case

Response 9: We sincerely appreciate the reviewer’s comment, and the line has been deleted, and the paragraph edited accordingly.